# An Adaptive Strategy for Tuning Duplicate Trails in SAT Solvers

**Wenjing Chang [1,2,\*], Yang Xu [2,3] and Shuwei Chen [2,3]**

1   School of Information Science and Technology, Southwest Jiaotong University, Chengdu 610031, China
2   National-Local Joint Engineering Laboratory of System Credibility Automatic Verification, Chengdu 610031, China; xuyang@home.swjtu.edu.cn (Y.X.); swchen@swjtu.edu.cn (S.C.)
3   School of Mathematics, Southwest Jiaotong University, Chengdu 610031, China
*   Correspondence: wenjing1021@163.com; Tel.: +86-182-8011-7952

**Abstract:** In mainstream conflict driven clause learning (CDCL) solvers, because of frequent restarts and phase saving, there exists a large proportion of duplicate assignment trails before and after restarts, resulting in unnecessary time wastage during solving. This paper proposes a new strategy—identifying those duplicate assignments trails and dealing with them by changing the sort order. This approach's performance is compared with that of the Luby static restart scheme and a dynamic Glucose-restart strategy. We show that the number of solved instances is increased by 3.2% and 4.6%. We also make a compassion with the MapleCOMSPS solver by testing against application benchmarks from the SAT Competitions 2015 to 2017. These empirical results provide further evidence of the benefits of the proposed heuristic, having the advantage of managing duplicate assignments trails and choosing appropriate decision variables adaptively.

**Keywords:** satisfiability problem; decision variable; restart; duplicate trails

## 1. Introduction

The Boolean satisfiability (SAT) problem has theoretical importance, which was proven to be Non-deterministic Polynomial complete (NP-complete) [1]. Since all NP-complete problems can be transformed into SAT problems in polynomial time, all NP-complete problems can be transformed into SAT problems in polynomial time. The SAT problem can be regarded as the "seed" of the NP-complete problem. If an efficient algorithm can be found to solve the SAT problem, other NP-complete problems can also be solved efficiently. The study of the SAT problem not only plays an important role in NP-complete theory, but also becomes a research hotspot because of its universality in practical applications. Lots of real-world problems, deriving from Artificial Intelligence (AI) planning [2,3], computer science [4,5], and automatic test pattern generation [6], can be formalized as a conjunctive normal form (CNF). A CNF formula is a conjunction of clauses, and each clause is a disjunction of variables; a variable, $x$, can be assigned as 0 (false) or 1 (true). A clause is satisfiable when at least one variable of this clause is assigned as being true. The SAT problem decides whether a set of variable assignments exist that make all clauses of the given CNF formula is satisfiable or not, or proves that any assignment of those variables cannot make the CNF formula satisfiable. Owing to the SAT problem being applied extensively in various practical domains, SAT problem solvers have gained remarkable improvements over the past decade. Lots of SAT solvers are mainly based on complete and incomplete algorithms. The complete algorithms cannot only get the solution of the SAT problem if it is satisfiable, but also provide a complete proof when there is no solution to the problem and prove the problem is unsatisfiable. Although an incomplete algorithm is especially salient for targeting a satisfiable random SAT formula, it is unable to prove unsatisfiability. Additionally, many practical application problems

generating from real-world utilities need to be proven regarding their unsatisfiability. Consequently, this paper mainly introduces the complete algorithms.

The Davis-Putnam-Logemann-Loveland (DPLL) algorithm [7] was proposed in 1962, which primarily employed the unit propagation rule, pure-literal rule, and split rule to search the solution space by a depth-first search. However, due to the particularity of the SAT problem, the DPLL algorithm has an exponential time complexity in the worst case. The international SAT competitions organized by research institutions around the world were first held in 1996, and the 22nd competition will be held in Lisbon. The SAT competitions have greatly boosted the solving efficiency of SAT solvers, and have also promoted the wider application of SAT problems in practice. At present, most of the state-of-the-art SAT complete algorithms are derived from DPLL algorithm, the most important of which is the conflict driven clause learning (CDCL) algorithm [8], an extension of the DPLL algorithm, by involved effective search techniques, such as the learnt clause scheme, heuristic decision strategy, restart techniques, and lazy data structures. In 1996, the GRASP solver [8], which is the prototype of the CDCL algorithm, introduced non-synchronous backtracking and a learnt clause scheme, and reduced the searching space to a large extent. In 1997, the head/tail list data structure was employed in the SATO solver [9], and vastly enhanced the efficiency of Boolean constraint propagation (BCP). In 2001, the Chaff solver [10] was integrated into the watched-literal data structure and the low-cost variable state independent decaying sum (VSIDS) decision strategy. In 2002, the BerkMin solver [11] added a learned clause database deletion strategy to avoid memory explosion because of an increasing number of learnt clauses. In 2004, the Zchaff solver [12] improved the implementation of programming based on the Chaff solver, further increasing the solution efficiency. In 2005, the Minisat solver [13], with a strong capability of the search space by optimizing the code structure, and a majority of solvers that entered the SAT competition later were based on Minisat. In 2009, the author of the Glucose solver [14] proposed a new learnt clause reduction method based on the literals blocks distance (LBD), and a dynamic restart strategy, the Glucose solver, got the first prize of the application group in the 2011 SAT competition. The Lingeling solver [15], which was also based on the CDCL algorithm, respectively won the 2013 and 2014 SAT competition about the application group. In 2016, Liang Jia Hui et al. designed the MapleCOMSPS solver [16,17], which utilized a new decision branching method—learning rate branching (LRB)—and got the first prize of the main-track group in the 2016 SAT competition and the second prize of the main-track group in 2017, respectively. In 2017, the solver, Maple_LCM_Dist [18], got the first prize of the main-track group in the 2017 SAT competition, which used a learnt clause minimization approach, and lots of competing solvers were based on it in the 2018 SAT competition.

On the one hand, one of the most surprising aspects of the relatively recent practical progress of CDCL solvers is the decision branching selection heuristics. Different branches lead to different search paths, which further affects the solution time. CDCL solvers select decision variables based on a strong activity heuristic. Initially, the score of each variable is the frequency of a literal occurrence in all clauses, and the score of variables that are involved in conflicts is increased additively by 1. These scores are sorted in a trail. More significantly, the size of the real-world instances always have many thousands, and even millions, of clauses and variables. Then, the score of each variable that occurs the most frequently is also high, so few variables of the front of the trail are still the same even if the trail updates scores periodically (every 256th conflict), because the increment is one, which is too small when compared with its initial score. Consequently, when solving an instance, which contains millions of clauses and variables, it is likely that there will be a large number of the same assignment trails.

On the other hand, a light-weight component caching [19]—phase saving—is conceived as a progress saving technique, which assigns the decision variable to the same phase that it was assigned before a restart. Additionally, it is obvious that a frequent restart [20] was introduced in current CDCL solvers, where it hit only a few new conflicts between two succeeding restarts. Therefore, due to the wide adoption of VSIDS and phase saving, a large proportion of the assignment trails are re-created exactly like the ones before the restart. It empirically exploits the observation that CDCL solvers tend

to make the same variables in a similar order: For example, in reference to [21,22], Van Der Tak et al. showed that often a large part of the trail can be reused after restarts and presented an algorithm for the detachment of repeated sequences to avoid assigning repeated variables; in [23], Jiang and Zhang observed that backtracking also exhibits a similar phenomenon and presented a partial backtracking strategy—this strategy no longer determines the backlevel according to the learnt clause, but directly back to the decision level where the non-repeating variables are located. Nonetheless, all these methods focused on reducing the computational costs of performing a restart, thus the occurrence of a large number of duplicate assignment trails was not avoided, which resulted in the same solution space. In terms of this issue, different decision heuristics have different assignment trails and then lead to different search paths. Thus, some solvers integrated different decision strategies and tuned decision heuristics on the fly. In [24], Shacham and Yorav proposed an adaptive framework, which switched the decision strategies dynamically based on an estimate of how well the decision strategy is expected to solve an instance during a certain time. In [25], Siddiqi and Huang proposed a new decision framework that incorporated multiple decision strategies, by switching amongst them on the fly based upon the outcomes of their effectiveness during the search. However, those evaluative criterions for switching on the fly between decision heuristics in these above methods are complicated and inaccurate.

In summary, the paper first analyzes the universality of repeated variable assignment and then proposes a method to dynamically alter duplicate assignment trails only by changing the scores of variables. If the number of duplicate assignment sequences generated in the search process is greater than the setting threshold parameter, then the activity of the corresponding variables is increased; that is, the increasing activity of the variables is used to change the order of the variable assignment, and, accordingly, the search path is transformed. The remaining part of this paper is organized as follows. Section 2 introduces the framework of the CDCL algorithm and restart strategies that are widely employed in CDCL solvers over the years. Section 3 gives an example to illustrate this phenomenon of a repeated assignment trail and presents the implementation details of identifying duplicate trails and proposes a method of tuning the order of the variable assignment dynamically. Section 4 presents the experimental results, showing that the solving performance of the improved solver that utilizes the proposed strategy is enhanced. Finally, we draw conclusions and provide thinking about future research in Section 5.

## 2. CDCL Algorithm

The CDCL algorithm is predominantly an extension of the original DPLL algorithm.

### 2.1. CDCL Algorithm

Algorithm 1 presents the framework of the CDCL algorithm. Typically, the procedure of pickDecisionVar is to choose an unassigned decision variable. If all variables are assigned, the formula, $\Sigma$, is satisfiable; otherwise, the current level indicator (line 9) must be updated and the polarity must be confirmed by the select phase function (line 10). If there is a conflict in the unit propagation process (line 11), a learnt clause is derived by the conflict analysis function (line 12), mainly according to the first UIP (unique implication point) [26] learnt clause scheme, and the backtrack level, *btl*, is computed by the compute back level function (line 13). If the *btl* is equal to 0, the formula is unsatisfiable (line 14), otherwise, *learntclause* is added to the learnt clauses database, $\Delta$, and the *btl* level is backtracked to by the back jump function (line 19). Then, the current decision level to the *btl* (line 20) is further updated. When it is time to reduce the learnt clauses database, the reduce DB procedure is executed. Each component critically influences the performance of the solver.

---

**Algorithm 1. Typical CDCL Algorithm**

---

Input:   CNF formula Σ
Output: SAT or UNSAT
1   *ζ*=Φ   //*ζ* is the variable assignment set, Φ is an empty set
2   Δ=Φ   //Δ is the learnt clause database
3   *dl*=0   //*dl* is the decision level
4   **while** (*true*) **do**
5     *var*=pickDecisionVar(Σ)
6     **if** (*ζ*⊨Σ) **then**
7        **return** SAT
8     **else**
9        *dl*=*dl*+1
10        *ζ*=*ζ*∪{selectPhase(*var*)}
11        **while** (UnitPropagation(Σ, *ζ*)==conflict) **do**
12           *learntclause*=conflictAnalysis(Σ, *ζ*)
13           *btl*=computeBackLevel(*learntclause*, *ζ*)
14           **if** (*btl*==0) **then** return UNSAT
15           **end if**
16           Δ=Δ∪{ *learntclause* }
17              **if** (restart())   **then**   btl=0
18              **end if**
19              backJump(*btl*)
20              *dl*=*btl*
21        **end while**
22        **if** (timeToReduce())   **then**   reduceDB(Δ)
23        **end if**
24     **end if**
25   **end while**

---

## 2.2. Restart Strategies

In addition to the general process described above, the restart is also one of the most surprising aspects of the relatively recent practical progress of SAT solvers. It was first proposed to avoid heavy-tail behavior in a combinatorial search [27]. For the solving of hard instances, the solver may get stuck in a complex part of the search space (i.e., having explored thousands of branches), then restart is applied as a solution and the current search is stopped. Restart policy serves to move the solver to a different part of the search space. Many restart policies have been proposed from various points of view.

We primarily introduce those restart strategies that are commonly used in the state of art SAT solvers.

1.   Luby Restarts Scheme Item [28]

Luby's policy can be formally defined as:

$$t_i = \begin{cases} 2^{k-1}, if \ i = 2^k - 1 \\ t_{i-2^{k-1}+1}, if \ 2^{k-1} \le i \le 2^k - 1 \end{cases} \tag{1}$$

where, *t* represents the value of the sequence, *i* and *k* are integers and the range of values starts from 1. Hence, the Luby sequence is as follows: 1, 1, 2, 1, 1, 2, 4, 1, 1, 2, 4, 8, 1, 1, 2, 1, ..., and was proved as being universally logarithmically optimal for search algorithms in which the runtime distribution of the problem is unknown. Of course, in fact, the Luby sequence is multiplied by a constant, "unit run" (in general, between 32 and 256). For example, if we set a "unit run" in this sequence to be 32 conflicts,

then the actual restart intervals are: 32, 32, 64, 32, 32, 64, 128, ... MiniSAT 2.2 [13] defined a unit run to be 100 conflicts, meanwhile, the solver, Rsat [29], used a unit run of 512 conflicts. Since unit runs are commonly short, solvers exhibit frequent restarts by using the Luby restart strategy.

2.  Dynamic Restart Scheme

The dynamic restart is triggered in the case when certain conditions are met. For instance, in [14], Biere changed the restart frequency by measuring the agility of the search process, as the agility is related to the rate of the recently flipped assignment. In [30], Audemard et al. suggested to postpone restarting when the solver appears to be close to finding a satisfying assignment, which is based on the local average trail size, when conflicts happen, being extremely greater than the global one; in other words, numerous variables exist that are assigned suddenly and unusually. It is a very aggressive restart strategy. In addition, this strategy was employed in Glucose solvers, named the Glucose-restart strategy. In [31], Biere et al. presented a new restart method by computing the simple moving average (SMA) value and cumulative moving average (CMA) value; if SMA > c*CMA (c > 1), then a restart was triggered. In [32], Nejati et al. proposed a multi-armed bandit restart policy, which adaptively switches between different restart policies.

## 3. Tuning Duplicate Trails Method

### 3.1. Motivation

Consider the example formula, $F$, and the activities of variables before the restart shown in Table 1.

$$F = \neg x_1 \vee x_2 \vee \neg x_6, \neg x_1 \vee \neg x_5, x_1 \vee x_4, \neg x_2 \vee \neg x_7 \vee \neg x_9, \neg x_2 \vee x_5 \vee x_6, x_3 \vee \neg x_4 \vee \neg x_7, \neg x_3 \vee x_8, \neg x_3 \vee \neg x_7 \vee x_9$$

Since the VSIDS decision strategy increased the activity of variable that was involved in the conflict analysis, Table 2 shows the activities scores of variables after the restart and constant conflicts. From Table 3, we obtained the resulting assignment trail before the restart; "$x_i \rightarrow x_j$" means that variable $x_i$ is a decision variable, and variable $x_j$ is an implied variable. Then, the conflict occurs when $x_7$ is propagated at level 5. Due to first-UIP scheme, we obtained the learnt clause, $\neg x_2 \vee \neg x_3 \vee \neg x_7$.

**Table 1.** VSIDS scores before the restart.

| Variable | $x_1$ | $x_2$ | $x_3$ | $x_4$ | $x_5$ | $x_6$ | $x_7$ | $x_8$ | $x_9$ |
|---|---|---|---|---|---|---|---|---|---|
| activity score | 89.5 | **70.7** | **59.6** | 65.3 | 80.3 | 77.6 | **55.2** | 57.5 | 45.6 |
| phase | True | True | True | True | False | True | True | True | **False** |

**Table 2.** VSIDS scores after the restart.

| Variable | $x_1$ | $x_2$ | $x_3$ | $x_4$ | $x_5$ | $x_6$ | $x_7$ | $x_8$ | $x_9$ |
|---|---|---|---|---|---|---|---|---|---|
| activity score | 89.5 | **79.8** | **61.4** | 65.3 | 80.3 | 77.6 | **58.6** | 57.5 | 46.8 |
| phase | True | True | True | True | False | True | True | True | **True** |

**Table 3.** Assignment trails before and after the restart.

| Decision Level | 1 | 2 | 3 | 4 | 5 |
|---|---|---|---|---|---|
| trail before restart | $x_1 \rightarrow \neg x_5$ | $x_6 \rightarrow x_2$ | $x_4$ | $x_3 \rightarrow x_8$ | $x_7 \rightarrow \neg x_9$ |
| trail after restart | $x_1 \rightarrow \neg x_5$ | $x_2 \rightarrow x_6$ | $x_4$ | $x_3 \rightarrow x_8, x_7$ | $x_9$ |

From Tables 1–3, we can easily conclude that (1) CDCL solvers will favor more frequent restarts, causing only a few clauses to be learned between two restarts, and, consequently, the activity scores of some variables will change slightly because of the frequent restarts; (2) these variables are repeatedly assigned in the new trail, which is ensured by the phase-saving heuristic used by most CDCL solvers.

Let us compare the assignment trails before and after the restart. As shown in Table 3, the first seven assigned variables in both trails remain the same, albeit in a different order, namely, $\{x_1, x_5, x_2, x_6, x_4, x_3, x_8\}$. As previously mentioned, after restarting, the solver tends to reassign many of the assignment trails; that is to say, the assignment trail is duplicated. To clarify this phenomenon intuitively, we applied the solver, Glucose3.0, to solve an instance: Aaai10-planning-ipc5-pathways-13-step17.cnf, which has been randomly selected from the SAT Competition 2014 Application benchmark, by adopting the Luby restarts strategy and dynamic Glucose-restart strategy, respectively. From Figure 1, the $x$-axis represents the numbers of restarts, unlike restart schemes that generate different restart numbers. Meanwhile, the $y$-axis represents the number of the duplicate assignment trails (the denotation of *S.size()* will be discussed in Section 3.2). It is visually obvious that the overwhelming majority of duplicate assignment trails are made by the solver, regardless of the restart strategy adopted by the solver.

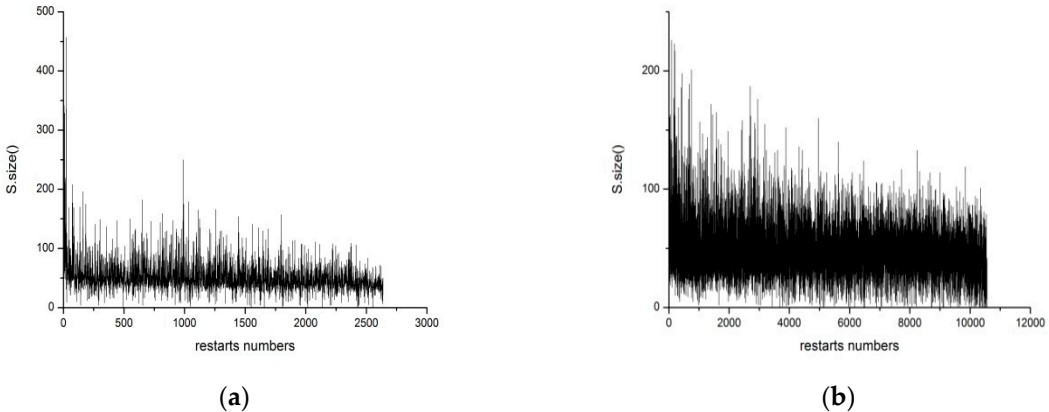

**(a)**            **(b)**

**Figure 1.** The duplicate assignment trails of: (**a**) The Luby restart; and (**b**) Glucose-restart.

### 3.2. Duplicate Trails

Referring to the example in Section 3.1., the variable, $x_1$, which still has the highest VSIDS score after the restart, is chosen as the decision variable and is assigned as true again at level 1. Consequently, this will cause an implied literal $x_5$, and it is thus the same before and after the restart. Note that the second decision literal after the restart is $x_2$ and its implied literal is $x_6$, and it no longer matches the trail before the restart, where the decision literal is $x_6$ and the corresponding implied literal is $x_2$. In addition, the third decision literal and the fourth decision literal are also the same in both trails (between the two trails). Here, there are two literal assignment trails in the first four levels before and after the restart, that is $S_{before} = \{x_1, \neg x_5, x_2, x_6, x_4, x_3, x_8\}$ and $S_{after} = \{x_1, \neg x_5, x_6, x_2, x_4, x_3, x_8, x_7\}$. The two search space, which is determined by the sequence, $S_{before}$ and $S_{after}$, are the same since the search path according to $S_{before}$ is: $x_1 = 1 \wedge x_5 = 0 \wedge x_2 = 1 \wedge x_6 = 1 \wedge x_4 = 1 \wedge x_3 = 1 \wedge x_8 = 1$, equivalent to $x_1 = 1 \wedge x_5 = 0 \wedge x_6 = 1 \wedge x_2 = 1 \wedge x_4 = 1 \wedge x_3 = 1 \wedge x_8 = 1$, which is the search path of $S_{after}$. Therefore, we define a sequence, $S$, containing those variables (decision variables and propagation variables) that keeps the reduced formula the same before and after the restart, in the above example, as S = $\{x_1, x_5, x_2, x_6, x_4, x_3, x_8\}$. Consequently, we checked whether the assignment sequence, $S$, is the same one as when each restart occurs. Noticeably, *S.size()* stands for the number of duplicate assignment variables. In the above example, *S.size()* = 7. The greater the value of *S.size()*, the greater the number of matching variables. When the value of *S.size()* remains large over a period of time, it indicates that the search path remains almost constant during this time. Aiming for this situation, the search path should be tuned by a dissimilar decision variable. Here is the sketch of the algorithm of the identification of duplicate trails.

In algorithm 2, suppose $x_{next}$ is the next assigned variable, *dl* is the current decision level, *decisionLevel* is the decision level before the restart, trail_order[] presents the variable assignment trail, and trail_order[*dl*] presents the decision variable of the decision level, *dl*. The assignment sequence is checked to ascertain whether it is the same after each restart or not; that is, deciding whether those

variables, which are ahead of $x_{next}$ in the assignment sequence before the restart, are the same as those variables, which are ahead of $x_{next}$ in the assignment sequence after the restart. For another, those variables are sorted by activity in the assignment trail, so only the activity[trail_order[$dl$]] value and activity[$x_{next}$] value need to be judged. If the activity[$x_{next}$] is greater than the activity[trail_order[$dl$]], which shows existing repeated variables, then *S.size* should be increased. When *S.size* meets certain conditions, namely, MIN<= *S.size* <= MAX, then the addition of counter *count* is needed. If the value of *count* is greater than a parameter *threshold*, it indicates that the assignment sequence is always repeated, and the changeTrailOrder() function is used to change the activity value ordering of the variable; if not, the pickDecisionVar function continues to be used. However, the question is how to set the parameter values of the MIN, MAX, and *threshold* for miscellaneous instances. Different parameters provide different behaviors. It is extremely tough to find a consistent ranking of solvers on the different sets of benchmarks. In the SAT community, experimental evidence is often required for any idea. We now show how we set the parameters of the MIN, MAX, and *threshold* by observing the performance of Glucose3.0 on the benchmarks, originally from the SAT 2015 Application benchmark only. Additionally, the CPU cutoff used was 1000 s. The Glucose3.0 solver is one of the award-winning SAT solvers in the corresponding competitive events since 2009. Table 4 supplies the number of solved instances when changing the parameters. The parameters, *MIN* = 20, *MAX* = 50, and *threshold* = 10, have the best performance compared to all the other parameters.

---

**Algorithm 2. Identification of duplicate trails**

---

Input: variable score array *activity[]*
Output: call changeTrailOrder() or PickDecisionVar()
1　　*count*←0
2　**for** dl←1 to *decisionLevel*
3　　　**if** activity[trail_order[*dl*]]< activity[$x_{next}$]
4　　　　　**then**　　S.size++
5　　　　**end if**
6　**end for**
7　**if** MIN<=S.*size*<=MAX
8　　　**then** count++
9　**end if**
10　　**if** count>threshold
11　　　　**then** changeTrailOrder()
12　**else**
13　　　pickDecisionVar()
14　**end if**

---

**Table 4.** Number of solved instances with different parameter values.

| MIN | MAX | *Threshold* | Sum |
|---|---|---|---|
| 10 | 30 | 5 | 237 |
| 20 | 40 | 10 | 241 |
| 30 | 50 | 15 | 240 |
| 20 | 50 | 10 | **244** |
| 10 | 40 | 15 | 239 |

The purpose of the changeTrailOrder() function is to change the sort order of variables in the trail, in essence, it is changing the activity of each variable. Certain branching heuristic strategies, which are adopted in modern CDCL solvers, increase the activity of those corresponding variables only by 1. The more the variable is associated with constructing the conflict, the larger its activity. In the subsequent process, variables with large activity values are preferred. Thus, it is believed that the easier it is to avoid conflicts that have occurred before, the easier it is to reduce the search space.

However, due to restarts that are frequently used in modern SAT solvers leading to a similar learning process, this further leads to the similarity of a learnt clause and may cause the changing of activities of a few variables on this occasion, thus further generating the repeated assignment sequence. Therefore, to change the activity value of variable to a large extent, the additional bonus value is added according to the number of times that a variable is responsible for conflict analysis. If the number of times the variable participates in the conflict analysis is greater, the earlier the variable is selected for unit propagation, and the more likely the conflict will occur. Between the two restart intervals, the number of times each variable has participated in the conflict analysis process is counted, and the variable that has the maximum count value is recorded. When the changeTrailOrder() function is used, the activity of the variable that has the maximum count is increased by 100, with the purpose of tuning the sequence of the variables substantially.

We conducted a comparison experiment by implementing the proposed algorithm 2 as the part of Glucose3.0, using two kinds of different restart policies for each to solve the aaai10-planning-ipc5-pathways-13-step17.cnf instance.

From Figure 2, it is evident that plenty of S.size() = 0 exists. Apparently, S.size() is equal to zero, which shows that the assignment trail did not contain a repeated variable and searched from a diverse path, hence it reveals that changing the order of the variables has a role in choosing a different path, and further illustrates the effectiveness of this algorithm. Additionally, from Figure 1b, we can see that the value of S.size() is approximately between 20 and 80. Meanwhile, after utilizing the proposed method, it is shown that the value of S.size() is between 0 and 20. This kind of phenomenon, on the one hand, means that the reduction of the value of S.size() implies that the number of duplicate assignment trails is decreasing. On the other hand, it further indicates that the proposed strategy is workable, due to the fact that it dynamically tunes decision variables and hence explores different branches.

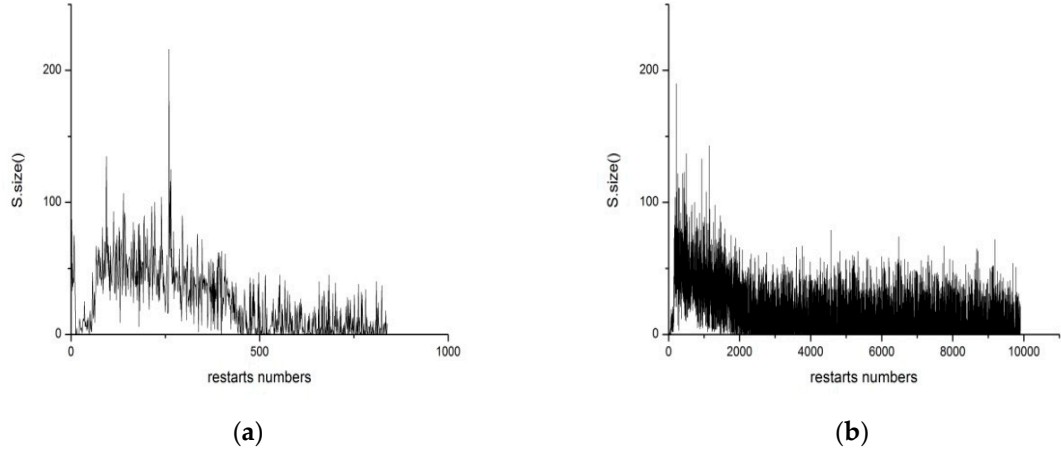

**(a)**        **(b)**

**Figure 2.** The duplicate assignment trails of: (**a**) Luby restart using the proposed scheme; (**b**) Glucose-restart using the proposed scheme.

## 4. Experimental Results

To better demonstrate the performance of the proposed strategy, we quote two of the leading SAT solver developers, Professors Audemard and Simon:

> *We must also say, as a preliminary, that improving SAT solvers is often a cruel world. To give an idea, improving a solver by solving at least ten more instances (on a fixed set of benchmarks of a competition) is generally showing a critical new feature. In general, the winner of a competition is decided based on a couple of additional solved benchmarks.*

In this section, we complete four solvers: Glucose_Luby, Glucose_Luby-IDT, Glucose_dyn, and Glucose_dyn-IDT. All four solvers are based on the Glucose solver3.0 (http://www.labri.fr/perso/

lsimon/glucose/). The Glucose_Luby solver implements the Luby static restart strategy, and the Glucose_Luby-IDT solver uses the proposed identification duplicate trails strategy based on the Glucose_Luby solver; similarly, the Glucose_dyn solver implements the dynamic Glucose-restart strategy, and the Glucose_dyn-IDT solver is a modified version solver configured with our proposed method, which is based on the Glucose_dyn solver. In the evaluation, we use the following parameters: MIN = 20, MAX = 50, *threshold* = 10, as shown in Table 4. The experiment was implemented on a 64-bit machine with 8 Gb of memory and an Intel Core i3-3240 CPU 3.40GHz processor. The benchmarks for the experiments originate from the SAT competitions, organized since 2002, which objectively compare the performances of a wide variety of SAT solvers, and have been an effective driving force for SAT solver development. We ran these four solvers on all the application benchmarks (obtained from a diverse set of applications) from the SAT competitions of 2015, 2016, and 2017. For each instance, the solver was allocated 3600 s of CPU time. Table 5 presents the number of solved instances by each solver.

**Table 5.** Number of solved instances by four solvers.

| Benchmarks | Status | Glucose_Luby | Glucose_Luby-IDT | Glucose_dyn | Glucose_dyn-IDT |
|---|---|---|---|---|---|
| | sat | 148 | 149 | 137 | 152 |
| Sat2015 (300) | unsat | 70 | 72 | 93 | 92 |
| | sum | 218 | 221 | 230 | 244 |
| | sat | 58 | 59 | 56 | 62 |
| Sat2016 (300) | unsat | 62 | 66 | 76 | 76 |
| | sum | 120 | 125 | 132 | 138 |
| | sat | 66 | 67 | 72 | 80 |
| Sat2017 (350) | unsat | 62 | 68 | 69 | 64 |
| | sum | 128 | 135 | 141 | 144 |
| | sat | 272 | 275 | 265 | 294 |
| Total (950) | unsat | 194 | 206 | 238 | 232 |
| | sum | 466 | **481** | 503 | **526** |

As shown in Table 5, the total number of solved instances by Glucose_Luby-IDT(Glucose_dyn-IDT), which incorporates our approach, is greater than that solved by Glucose_Luby(Glucose_dyn), respectively. Compared with the Glucose_Luby(Glucose_dyn) solver, the number of solved instances of Glucose_Luby-IDT(Glucose_dyn-IDT) is increased by 3.2% (4.6%), respectively. By comparison, the number of solved instances is mainly increased by solving UNSAT instances for the static restart strategy, and for the dynamic restart strategy, the number of solved instances is mainly enhanced by solving SAT instances. Besides, the dynamic restart policy is generally superior to the static policy for solving these instances.

Figure 3 shows the cactus plot comparing the performance comparison of these four solvers. Figure 4 displays the performance comparison of four solvers over SAT instances and UNSAT instances, respectively. The $x$-axis denotes the numbers of solved instances while the $y$-axis denotes the time required to solve them. The line is farther towards the right, meaning that more instances are solved by the corresponding solver. Meanwhile, the lower the line, the faster the corresponding solver works.

As can be seen from the cactus plot in Figure 3, the cactus of Glucose_dyn-IDT (Glucose_Luby-IDT) is always to the right and below the curve of Glucose_dyn(Glucose_Luby), respectively. Additionally, the cactus of Glucose_dyn-IDT is on the far right and at the bottom. It states that the Glucose_dyn-IDT solver has the best solving performance. From Figure 4a, for the SAT instances, the performance of the Glucose_dyn-IDT solver is the best, and the solution performance of the Glucose_Luby solver and the Glucose_Luby-IDT solver is similar; from Figure 4b, for the UNSAT instances, the performance of the four solvers varies greatly, and the Glucose_dyn solver performs the best.

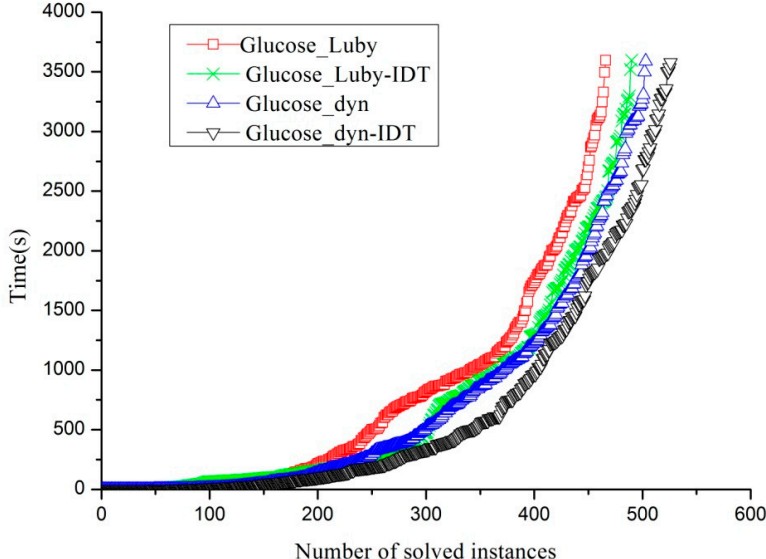

**Figure 3.** Performance of four solvers over 950 instances

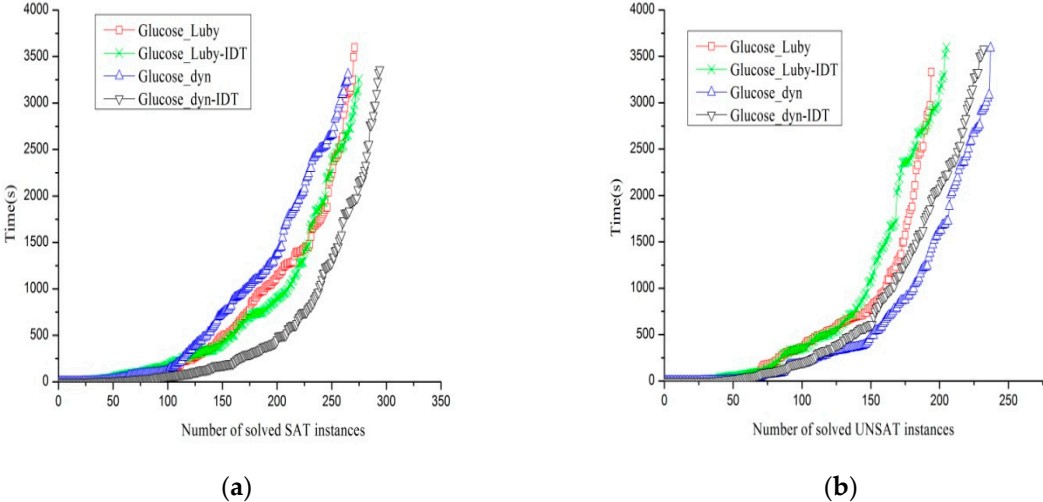

(**a**)                                         (**b**)

**Figure 4.** Performance of four solvers over: (**a**) SAT instances; (**b**) UNSAT instances.

The above experimental results show that the proposed strategy can work well either for the Luby static restart or the dynamic Glucose-restart policy. To better illustrate the advantages of the proposed method, we further compared it with the MapleCOMSPS solver, which took first place in the main-track group of the SAT 2016 competition. The MapleCOMSPS is a modification of the COMiniSatPS solver [33], with slight changes to the VSIDS. The MapleCOMSPS_IDT solver implements this proposed method founded on MapleCOMSPS. We ran these two solvers on all the application benchmarks (obtained from a diverse set of applications) from the SAT competitions of 2015, 2016, 2017, and 2018. For each instance, the solver was allocated 3600 s of CPU time. Table 6 presents the number of solved instances by the MapleCOMSPS and MapleCOMSPS_IDT solver.

From Table 6, the total number of solved instances by MapleCOMSPS_IDT increases by 1.7% compared with the MapleCOMSPS solver. The cactus plot comparing the performances of the two solvers can be observed in Figure 5. Although the amount of growth is small, the overall solution performance is improved slightly by the modified MapleCOMSPS_IDT solver.

Figure 6 shows the performance comparison of the MapleCOMSPS vis-a-vis the MapleCOMSPS_IDT over the SAT instances and UNSAT instances, respectively. It can be seen clearly that our proposed strategy has a better performance for the SAT instances and UNSAT instances.

**Table 6.** Number of solved instances by the two solvers.

| Benchmarks | Status | MapleCOMSPS | MapleCOMSPS_IDT |
|---|---|---|---|
| | sat | 154 | 155 |
| Sat2015(300) | unsat | 102 | 101 |
| | sum | 256 | 256 |
| | sat | 65 | 78 |
| Sat2016(300) | unsat | 78 | 66 |
| | sum | 143 | 144 |
| | sat | 90 | 86 |
| Sat2017(350) | unsat | 68 | 76 |
| | sum | 158 | 162 |
| | sat | 123 | 110 |
| Sat2018(400) | unsat | 89 | 110 |
| | sum | 212 | 220 |
| | sat | 432 | 429 |
| Total(1350) | unsat | 337 | 353 |
| | sum | 769 | 782 |

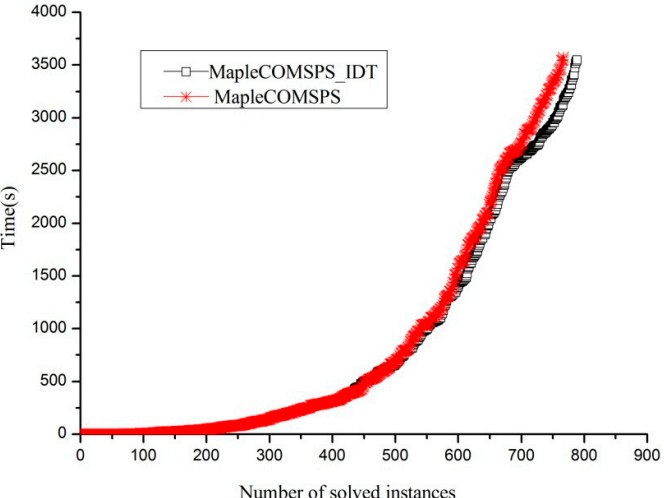

**Figure 5.** Performance of MapleCOMSPS_IDT vs. MapleCOMSPS over 1350 instances.

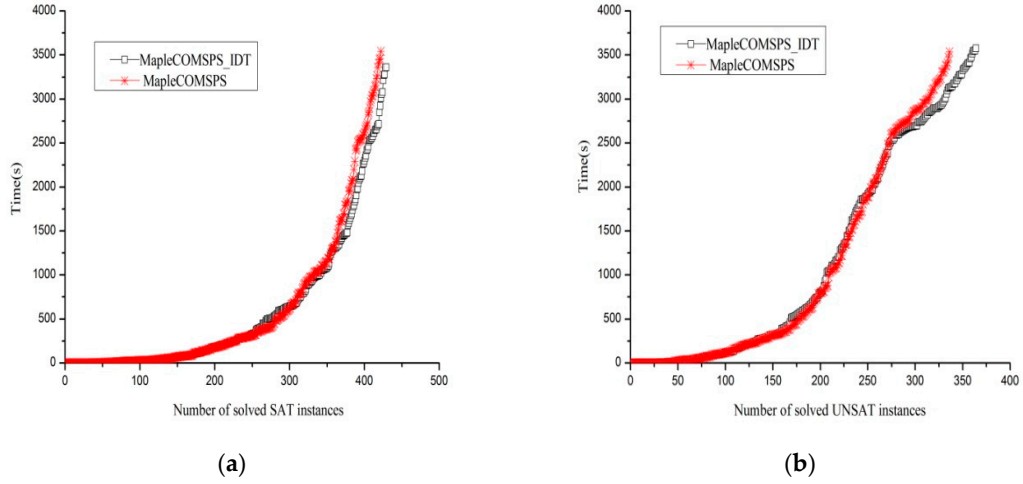

(**a**)          (**b**)

**Figure 6.** Performance of MapleCOMSPS_IDT vs. MapleCOMSPS over: (**a**) SAT instances; (**b**) UNSAT instances.

## 5. Conclusions

We proposed a new strategy for the identification of duplicate assignment trails after frequent restarts and dealt with them by changing the sort order adaptively. By detecting the sequence of the assignment variables generated in the search process, if the number of repeated sequences is greater than the set threshold, the activity of the corresponding variables is increased, and the assignment order of the variables is changed to tune the search path. The experimental results showed that the proposed algorithm adaptively changes the order of the variables' assignment and reduces the duplicate paths to a certain extent. At the same time, we implemented the proposed strategy on Glucose and MapleCOMSPS, respectively. The experimental results showed that our modified solvers have a better performance regarding the solving of the application benchmarks from the SAT competitions of 2015 to 2018.

**Author Contributions:** Conceptualization, W.C.; methodology, W.C.; validation, S.C. and Y.X.; writing—original draft preparation, W.C.; writing—review and editing, W.C.; supervision, Y.X.; funding acquisition, Y.X.

**Funding:** This research was funded by the National Natural Science Foundation of China (Grant No. 61673320), and by the Fundamental Research Funds for the Central Universities (Grant No. 2682018ZT10, 2682018CX59, 2682018ZT25).

**Acknowledgments:** The author would like to express their sincere appreciation to the editor and the anonymous referees for their valuable comments and suggestions.

**Conflicts of Interest:** The authors declare no conflict of interest.

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
