# Peer review of "An Adaptive Strategy for Tuning Duplicate Trails in SAT Solvers"

_symmetry, doi:10.3390/sym11020197_

Round 1

Reviewer 1 Report

In the paper, authors presented a new strategy-identifying duplicate assignments trails using sorting idea.

Small issues:

-before citing should be a space, that is instead of example[1], there should be -- example [1],

-the quote in sec 4 should be deleted,

-the charts are illegible. Can you enlarge them and add colors?

Major issues:

-Introduction should be rewritten. The authors show works from a few years ago and they do not mention anything about the latest achievements. 90% of citations older than 4 years should be removed and replaced with newer ones.

-Was this problem solved by artificial intelligence? Please provide correct references. It is worth noting the latest developments in this field

-Sections with proposed ideas like second one, should be rewritten. Please write the idea with all mathematical formulas, than add pseudocode. Right now, there is pseudocod and explanations of it. It is wrong approach, it should be full mathematical model, than pseudocode and some remarks to pseudocode. 

-In Algorithm 1 is incomprehensible. The authors use the function and then describe it in the text (the article is more documentation? it should be rewritten). Moreover, the explanation is not understandable.

-equations should be numbered and explained. For example, the equation in section 2.2 is not readable, which means the variable k?

-the equations are poorly formatted and unreadable.

-in the experiments section, the authors use different algorithms like Glucose_Luby-IDT, where is their source? Please, add a citation and explain the differences in the operation of algorithms in earlier sections.

- In summary, please do not focus on your future work, and describe extensively what has been done by the authors and how it will affect the field

-What is the computational complexity of the proposed method? The authors should add its full calculation.

Author Response

Small issues:

Point 1:before citing should be a space, that is instead of example[1], there should be example [1],

Response 1: I have modified it.

Point 2:the quote in sec 4 should be deleted,

Response 2: I have deleted it;

Point 3:the charts are illegible. Can you enlarge them and add colors?

Response 3:I have enlarge the charts and add colors.

Major issues:

Point 1:Introduction should be rewritten. The authors show works from a few years ago and they do not mention anything about the latest achievements. 90% of citations older than 4 years should be removed and replaced with newer ones.

Response 1: In introduction, according to the time sequence of SAT competition, some famous solvers in SAT competition are listed, and the development process of SAT solver is explained. In addition, I have added the famous solver in SAT 2017 competition and SAT 2018 competition.

Point 2:Was this problem solved by artificial intelligence? Please provide correct references. It is worth noting the latest developments in this field

Response 2: It means that problems in the field of artificial intelligence can be transformed into SAT problems for solving. And I have provided correct references.

 Point 3:Sections with proposed ideas like second one, should be rewritten. Please write the idea with all mathematical formulas, than add pseudocode. Right now, there is pseudocod and explanations of it. It is wrong approach, it should be full mathematical model, than pseudocode and some remarks to pseudocode. 

Response 3: in references, for example [16,23,32], the  proposed ideas are explained by pseudocod, so I don’t exactly how to explain the algorithm by full mathematical model.

Point 4:In Algorithm 1 is incomprehensible. The authors use the function and then describe it in the text (the article is more documentation? it should be rewritten). Moreover, the explanation is not understandable.

Response 4:  I have rewritten the Algorithm 1 and the explanation.

Point 5:equations should be numbered and explained. For example, the equation in section 2.2 is not readable, which means the variable k?

Response 5: I have modified it.

Point 6:the equations are poorly formatted and unreadable.

Response 6: In reference[28], the equations is written like this.

Point 7:in the experiments section, the authors use different algorithms like Glucose_Luby-IDT, where is their source? Please, add a citation and explain the differences in the operation of algorithms in earlier sections.

Response 7: I have added the citation, and explained the differences.

Point 8:In summary, please do not focus on your future work, and describe extensively what has been done by the authors and how it will affect the field

Response 8: I have deleted the future work in summary.

Point 9:What is the computational complexity of the proposed method? The authors should add its full calculation.

Response 9: The algorithm for solving SAT problem is NP-complete. The proposed method just improved the performance of solving SAT instances.

Reviewer 2 Report

This paper presents a novel strategy for identifying duplicate assignment trails after frequent restarts. The paper is well written and organized. The topic addressed is timely and in the scope of the journal, the methodology followed is well described and sounds ok. The experiments are well designed and analyzed.

Content suggestions and comments:

-Algorithm 1, line 1-2: what is phi? do you mean empty set?

-Algorithm 1, line 18-20: please, recheck the identation of these lines

-line 165: equation numbering is missing

-line 165: what are k, i, t, etc…?

-line 211: “it is visually obvious” -> it would better to find a quantitative explanation for this statement

-Algorithm 2: please, recheck the identitation of lines (for example: end for)

Author Response

Point 1:Algorithm 1, line 1-2: what is phi? do you mean empty set?

Response 1: Ф is an empty set, I have added it.

Point 2:Algorithm 1, line 18-20: please, recheck the identation of these lines

Response 2: I have rewritten the Algorithm 1

Point 3:line 165: equation numbering is missing

Response 3: I have added the number.

Point 4:line 165: what are k, i, t, etc…?

Response 4: I have added it.

Point 5:line 211: “it is visually obvious” -> it would better to find a quantitative explanation for this statement

Response 5: In Figure 1, the x-axis represents the numbers of restarts, the y-axis represents the number of the duplicate assignment trails. So from the figure, “it is visually obvious” that restart strategy make majority of duplicate assignment trails.

Point 6:Algorithm 2: please, recheck the identitation of lines (for example: end for)

Response 6: I have rechecked the algorithm 2, and the location of “end for” is right.

Reviewer 3 Report

The present paper considers the topic of SAT solvers and proposes a new method for the dynamic alteration of duplicate assignment trails.

In my opinion, while the results presented are of some interest, the paper suffers from low readability and would profit from more exhaustive explanations provided throughout all the sections.

To this end, I think that the Authors should:

1) enlarge the introduction, providing a more exhaustive introduction to SAT problems, in order to provide more elements also for readers who are not familiar with the problem;
2) provide a larger review of related works, better highlighting how they relate to the present work;
3) more clearly state the original contributions of the present work;
4) more mathematically rigorously and clearly describe the algorithmic original contributions that are proposed,

5) (last but not least) enlarge the computational section, in particular by widening the computational tests in terms of number of instances considered.

Author Response

Point 1:  enlarge the introduction, providing a more exhaustive introduction to SAT problems, in order to provide more elements also for readers who are not familiar with the problem;

Response 1: I have enlarged the introduction.

Point 2:  provide a larger review of related works, better highlighting how they relate to the present work;

Response 2: I have added one reference[16] about related works.

Point 3:   more clearly state the original contributions of the present work;

Response 3: I have clearly stated the original contributions of the present work.

Point 4:   more mathematically rigorously and clearly describe the algorithmic original contributions that are proposed,

Response 4: I think it is difficult to describe the advantages of the algorithm from a mathematical point of view. This paper has demonstrated the advantages of the algorithm through experiments.

Point 5:  (last but not least) enlarge the computational section, in particular by widening the computational tests in terms of number of instances considered.

Response 5: I have enlarged the computational section.

Round 2

Reviewer 1 Report

The authors did not understand all of my points, so I still think that the article should be improved.

Point 3:Sections with proposed ideas like second one, should be rewritten. Please write the idea with all mathematical formulas, than add pseudocode. Right now, there is pseudocod and explanations of it. It is wrong approach, it should be full mathematical model, than pseudocode and some remarks to pseudocode.

Response 3: in references, for example [16,23,32], the proposed ideas are explained by pseudocod, so I don’t exactly how to explain the algorithm by full mathematical model.

Comment: For example, the conflictAnalysis() function should have its own mathematical model, where the equations will be presented so that the reader knows exactly what is going on. This applies to the FULL model.

Point 6:the equations are poorly formatted and unreadable.

Response 6: In reference[28], the equations is written like this.

Comment: It was not corrected at all. Please see, for example line 239 on page 6 - it is unreadable. It must be corrected in whole paper. 

Point 8:In summary, please do not focus on your future work, and describe extensively what has been done by the authors and how it will affect the field

Response 8: I have deleted the future work in summary.

Comment: I wrote 'describe', not 'delete', here --  " describe extensively what has been done by the authors ".

Author Response

Point 3:Sections with proposed ideas like second one, should be rewritten. Please write the idea with all mathematical formulas, than add pseudocode. Right now, there is pseudocod and explanations of it. It is wrong approach, it should be full mathematical model, than pseudocode and some remarks to pseudocode.

Response 3: in references, for example [16,23,32], the proposed ideas are explained by pseudocod, so I don’t exactly how to explain the algorithm by full mathematical model.

Comment: For example, the conflictAnalysis() function should have its own mathematical model, where the equations will be presented so that the reader knows exactly what is going on. This applies to the FULL model.

ResponseThe main purpose of algorithm 1 is introducing the flow of CDCL algorithm. Each function, for example, unitPropagation(), pickDecisionVar(), conflictAnalysis(), has its own specific process. Due to the limitation of the length of the article, and the focus of this paper is the repetitive path, so this paper mainly introduced the flow of CDCL algorithm, which similar to Ref. [16,23,32] .

Point 6:the equations are poorly formatted and unreadable.

Response 6: In reference[28], the equations is written like this.

Comment: It was not corrected at all. Please see, for example line 239 on page 6 - it is unreadable. It must be corrected in whole paper. 

Response In line 239, “x1=1˄x5=0˄x2=1˄x6=1˄x4=1˄x3=1˄x8=1” isn’t an equation, it is the search path. In the background knowledge of SAT problems, “˄” means disjunction of variable, “x1=1” means the variable x1 is assigned 1(true), which had been introduced in the Section 1.

Point 8:In summary, please do not focus on your future work, and describe extensively what has been done by the authors and how it will affect the field

Response 8: I have deleted the future work in summary.

Comment: I wrote 'describe', not 'delete', here --  " describe extensively what has been done by the authors ".

ResponseI have described extensively what has been done by the authors.

Reviewer 3 Report

The authors have addressed my comments and the paper could be now be considered for possible acceptance.

Author Response

Thanks for the reviewer's comments

Round 3

Reviewer 1 Report

The authors did not take all my suggestions into consideration and because of the "limit" they are unable to do so. I leave the decision to the editor.

Author Response

Thanks for the reviewer's comments. 

Because  in the papers on SAT , such as Ref. [16,23,32], the framework of CDCL algorithm is briefly introduced. In case readers want to know the detailed process of each function in the CDCL algorithm, readers can refer to the code of Glucose solver, here is the website: http://www.labri.fr/perso/lsimon/glucose/